# Understanding Wood Surface Chemistry and Approaches to Modification: A Review

**DOI:** 10.3390/polym13152558

**Published:** 2021-07-31

**Authors:** Roger M. Rowell

**Affiliations:** Biological Systems Engineering, University of Wisconsin-Madison, Madison, WI 53558, USA; rmrowell@wisc.edu

**Keywords:** interface, interphase, absorption, adsorption, coatings, weathering, water repellency, mold, fungi, decay, hardness

## Abstract

Wood was designed, after millions of years of evolution, to perform in a wet environment. Nature is programmed to recycle it, in a timely way, back to the basic building blocks of carbon dioxide and water. All recycling chemistries start with an invasion of the wood surface. The surface of wood is porous, hygroscopic, viscoelastic, and anisotropic that is better defined in interface/interphase zones. This surface is dynamic and in constant change with changing humidity, temperature, oxygen levels, ultraviolet energy, microorganisms and stress. This chapter is a review of the chemical properties of a wood surface and performance issues associated with it.

## 1. Introduction

Wood has been used by humans for thousands of years as tools, fuel, weapons, structures and for recreation. We have used wood for so long that we have learned to design around the knowledge that it changes dimensions with changing moisture content, weathers, burns, and is degraded by a wide variety of micro and macro organisms.

There are many misconceptions about wood. First, that wood is renewable and sustainable. This is not true. Wood comes from trees so it is trees that are renewable and sustainable so we must put our emphasis on keeping our forests lands healthy. Second, wood was designed, by Nature, to be used as a building material. This is also not true. Nature is programmed to recycle wood in a timely way back to carbon dioxide and water using five basic chemistries [oxidation, reduction, dehydration, hydrolysis, free radical reactions]. Additionally, while we call wood a material, in a materials science definition of a material, it is not. The definition of a material is that its properties are uniform, consistent, predictable and reproducible. The properties of wood, however, vary from specie to specie, tree to tree, within the same tree and, even, within a board from the same tree. So, solid wood is a composite which is defined as two or more resources held together by some sort of mastic.

We have all gone to a lumber yard and bought wood to build a dog house, a piece of furniture or a deck. We know wood. We trust wood. We like the smell, look and feel of it. We know everything there is to know about wood. It is a common material used by common people. We would not think of making that dog house out of steel or ceramics or nano-tubes. We know that the dog house will not last forever but neither will the dog. We know wood to be temporary and we can build another one when the old one rots away.

The study of wood is best done by looking at it in differing levels of detail. There are several levels of details to consider: macro, sub-macro, micro, sub-micro, and molecular. We recognize wood at the macro level as a tree and this level can be broken down into two sub-categories, softwood and hardwoods. At the sub-macro level, we recognize wood as a solid board (in the rough) or as furniture, windows, doors, etc. At the micro level, we study the wood cell wall and identify different elements such as heartwood, sapwood, annual rings, etc. At the sub-micro level, we see cell structure such as lumens, pits, vessels, ray cells, etc. Finally, at the molecular level we can study the cell wall polymers (cellulose, lignin and hemicelluloses) and their building blocks of simple sugars, phenolic units as well as extractives structure and inorganic compounds.

## 2. Surface Chemistry

In the present case, we are looking at the surface chemistry of wood. There is a chemical surface and a physical surface. This chapter will deal with properties of surface chemistry and how that surface can be modified to improve properties and performance. Some properties of wood are controlled by surface chemistry and some by bulk chemistry.

To a wood chemist, it is easy to describe the surface of a piece of 404 stainless steel. To a chemist working with stainless steel, it is easy to describe the surface of wood but to a wood chemist, it is very hard to define the surface of wood.

A wood surface is a porous, hygroscopic, viscoelastic, anisotropic, three dimensional bio-polymer composite that is composed of cellulose, hemicellulose, lignin, extractives and inorganics. The surface is dynamic and in constant change with changing humidity, temperature, oxygen levels, ultraviolet energy, microorganisms and stress. 

How deep is a wood surface? (Figure 1). It has been shown that ultraviolet (UV) light penetrates a fir-wood surface to a depth greater than 70 µm, resulting in changes in the tensile strength of the fir-wood strips to a depth of 70 to 140 µm. [1]. So, we can use 70 to 140 µm as one definition of the depth of the interface surface of wood. The interface is the first few wood cells and then a transition to an interphase. There is no easy definition of the depth of the interphase and it varies depending on the application in question. For example, the interphase for the depth of penetration of a coating or an adhesive is different from that of the interphase depth for moisture sorption.

## 3. Property and Modification

### 3.1. Moisture Sorption

Because wood is hygroscopic, the interface/interphase of wood absorbs water as a vapor or liquid from the surrounding atmosphere. Moisture absorption starts in the interface and continues adsorbing moisture into the interphase until the fiber has reached the fiber saturation point (FSP) Figure 2. Loss of moisture results in cell wall shrinking.

According to the Dent sorption theory, water is added to the cell wall polymers in mono-layers [2]. Figure 3 shows the mechanism of water molecules adding to the cell wall. The hemicelluloses are the most hydroscopic polymer in the cell wall [3]. These polymers are also very accessible to moisture so may be the first site for moisture absoption. Water molecules enter the cell wall and start hydrogen bonding with other accessible hydroxyl groups. Moisture is sorbed either as primary water ● molecules and secondary water ○ molecules (Figure 3). Moisture opens the cell structure by “unzipping” hydrophilic polymer chains until the cell wall is fully saturated with bonded water [4]. Hydrogen bonds between hydroxyl groups on and between hemicelluloses, cellulose and lignin are constantly changing. As moisture is added to the cell wall, the fiber volume increases nearly proportionally to the volume of water added [5,6]. Swelling of the fiber continues until the cell matrix reaches the fiber saturation point (FSP) and water, beyond the FSP, is free water in the void structure and does not contribute to further swelling (Figure 3) [7]. This process is reversible, and fiber shrinks as it loses moisture below the FSP.

Sorption of moisture is much slower than desorption. That is, it is faster/easier to lose a molecule of water from the cell wall than it is to force another one in. A sorption isotherm (see Figure 4) is a plot of moisture content (M%) vs. relative vapor pressure (h = relative humidity/100 [8,9]. The difference between these curves is referred to as sorption hysteresis for wood [7]. The adsorbing curve is always lower than the desorbing curve and the A/D ratio generally ranges between 0.8 and 0.9 depending on the relative humidity and wood species [10].

### 3.2. Water Repellency

The terms water repellency and dimensional stability are often used interchangeably as if they were the same. They are very different concepts. Water repellency is a rate phenomenon and dimensional stability is an equilibrium phenomenon [11]. Confusion over these two concepts has led to some product failures in service costing contractors or owners considerable money.

A water repellent treatment is one that prevents or slows down the rate moisture or liquid water is taken up by the wood. Examples of water repellents include coating, surface applied oils or surface lumen filling. A dimensional stability treatment is one that reduces or prevents swelling in wood no matter how long it is in contact with moisture or liquid water. Examples of dimensional stability treatments include bulking the cell wall with polyethylene glycol, penetrating polymers or bonded cell wall chemicals, or cross-linking cell wall polymers [12].

Water repellent effectiveness (WRE) is measured as a time dependent function of increasing weight of liquid water that can penetrate the treated surface given:(1)$$\mathrm{WRE}=\frac{\mathrm{Wc}-\mathrm{Wt}}{\mathrm{Wc}}\times 100$$
where: Wc = Weight of water uptake) of control during exposure in water for “t” minutes Wt = Weight of water uptake) of treated specimen for the same “t” time.

As was stated before, water repellents are applied to wood principally to prevent or reduce the rate of liquid water flow into the surface cellular structure [13]. Moisture is physically blocked from entering lumens and penetration of the water must proceed by wicking through the cell wall. Moisture pickup can take a very long time moving through cell wall structure so this type of treatment can be confused as a treatment for dimensional stability.

Usually, the water repellent treatments involve the deposition of a thin layer of a hydrophobic substance onto external and to some extent, internal cell lumen surfaces of wood. The measured WRE varies between 0 and 100 percent depending on the time the test specimens are exposed to water. In some cases, the time to reach equilibrium may be weeks, months or even years but eventually, maximum swelling will be reached at equilibrium. A very effective surface water repellent treatment is shown in Figure 5. A drop of water on the treated surface does not soak in.

### 3.3. Weathering

Weathering is the general term used to define the surface degradation of wood exposed to the weather [14]. The degradation mechanism depends on a combination of factors found in nature: moisture, sunlight, heat/cold, chemicals, abrasion by windblown materials, and biological agents. The exterior of many houses has wood as the outermost barrier to the weather (siding, windows, decks, roofs, etc.). If we are to achieve long service life from these wood products, we must understand the weathering process and develop treatments to retard or stop this degradation. Failure to recognize the effects of weathering can lead to catastrophic failure of wood products. For example, if wood siding is left to weather for as little as 1–2 weeks before it is painted, the surface of the wood will degrade [15]. During this short exposure period, the surface of the wood will not appear different but photo oxidation has started. Application of paint after 1–2 weeks of weathering will not give a durable coating [16]. The surface of the wood has been degraded and it is not possible to form a good paint bond with the degraded surface. The paint will show signs of cracking and peeling within a few years. As the paint peels from the surface, the wood grain pattern can easily be seen on the back side of the paint. The peeling paint has lifted the damaged layer of wood from the sound wood underneath. The reasons for this will become apparent as we discuss the chemistry and degradation processes of wood weathering.

We see many examples of weathering. The gray roughened appearance of old barns, wood shake roofs, and driftwood are typical examples of weathered wood. Figure 6 shows the color change as the wood surface weathers.

In the absence of biological attack, weathering of wood can give a beautiful bright gray patina [17]. UV radiation has sufficient energy to chemically degrade wood structural components, mainly lignin. The wetting and drying of wood through precipitation and seasonal changes in relative humidity (RH), abrasion by windblown particulates, temperature changes, atmospheric pollution, oxygen, and human activities such as walking on decks, cleaning surfaces with cleaners and brighteners, sanding, and power-washing all contribute to the degradation of wood surfaces [18]. However, it is primarily the ultraviolet (UV) portion of the solar spectrum that initiates the process we refer to as weathering. It is a photo-oxidation or photochemical degradation of the surface [19]. The degradation starts immediately after the wood is exposed to sunlight [20]. First, the color changes, then the surface fibers loosen and erode, but the process is rather slow (Figure 7). It can take more than 100 years of weathering to decrease the thickness of a board by 5–6 mm. In addition to the slow erosion process, other processes also occur. As the lignin is degraded from the surface, it releases cellulose fibers and hemicelluloses polymers. The wood develops checks and raised grain [21].

Figure 7 shows the loss of surface fibers due to loss of lignin.

The loss of surface fibers results in a new surface and the UV degradation cycle continues. The loss of lignin in the cell wall can be seen in scanning electron micrographs Figure 8. The lignin content is higher in the middle lamella than in the cell wall, therefore the photo degradation occurs preferentially in this area of the wood surface. This is particularly noticeable in micrographs of a southern pine cross section before and after UV exposure.

There have been many studies to investigate the mechanism of wood weathering, and it has been clearly shown that the absorption of a UV photon can result in the formation of a free radical and that through the action of oxygen and water, a hydroperoxide is formed [22]. Both the free radical and hydroperoxide can initiate a series of chain scission reactions to degrade mainly lignin. On the basis of the depth of color change, degradation of wood as deep as 2500 µm following exposure of wood to weathering, however, this depth is beyond the limit for generation of free radicals. Today, most agree that weathering of wood is confined to the outer 25–300 µm.

In many weathering studies, weight loss, surface roughness, color changes, cracking, cupping, warping, and depth of erosion are measured [18]. For pine, the weight loss due to weathering erosion is 0.019%/h with an erosion rate of 0.121 μm/h. Latewood weathers much slower than spring wood giving rise to an uneven surface Figure 9.

### 3.4. Chemical Modification—Gas, Ketene, Cold Plasma

The wood surface contains hydroxyl groups on cellulose, hemicellulose and lignin. These can be used as bonding sites for modification chemistry to change properties and performance of wood products.

If a gas chemical system is used, the reaction will take place on the interface/interphase hydroxyl groups since gasses do not penetrate far into the wood structure.

The reaction with gaseous acetic anhydride results in esterification of the accessible hydroxyl groups in the cell wall with the formation of byproduct acetic acid [23].
WOOD-OH + CH_3_-C(=O)-O-C(C=O)-CH_3_ → WOOD-O-C(=O)-CH_3_ + CH_3_-C(=O)-OHAcetic Anhydride→Acetylated Wood Acetic Acid(2)

Acetylation is a single-site reaction which means that one surface acetyl group is on one hydroxyl group with no polymerization. This means that all of the weight gain in acetyl can be directly converted into units of surface hydroxyl groups blocked [24].

The byproduct acid can be eliminated if the reaction is carried out in ketene gas [24,25,26,27,28].
Wood–OH + CH_2_=C=O → Wood–O–C(=O)–CH_3_Ketene Acetylated Wood(3)

This chemistry occurs mainly on the wood surface due to poor penetration of the ketene gas.

Cold plasma chemistry can also be used to modify the surface of wood. [29,30,31,32,33]. The plasma used for wood modification is the same used for generating light in fluorescent lamps. The easiest way to technologically generate plasma is the electrical gas discharge. A voltage is applied in between two metal electrodes generating an electric field in the gaseous gap. One example of surface plasma uses hexamethyldisiloxane (HMDSO) which was deposited onto wood surfaces and investigated using electron spectroscopy for chemical analysis [32]. Plasma reactions were carried out in a stainless steel, parallel plate, cold plasma reactor. The presence of a crosslinked macromolecular structure, based on Si-O-Si and Si-O-C linkages was formed.

Pyrolysis mass spectroscopy was carried out to investigate the nature of the building blocks of the plasma generated macromolecular structure. Plasma modified samples exhibited very high water contact angle values (130 degrees) in comparison to the unmodified samples (15 degrees), indicating the presence of a hydrophobic surface. Figure 10 shows the surface of wood before and after plasma treatment. Atomic force microscopy images, collected both from unmodified and HMDSO-plasma modified samples, indicate the progressive growth of the plasma polymer resulting in the deposition of a smooth layer at 10 min treatment time [29].

### 3.5. Hardness

Surface hardness can be increased by the impregnation of acrylic dimers or larger molecules [34,35]. If a monomer is used, it will penetrate deep into the wood structure but larger molecules will only penetrate the first few cells in the wood surface. The reaction can be catalyzed using a vazo catalyst (2,2′-azobis-(2-methylbutyronitrile). Figure 11A shows a micrograph of an oak surface before reaction with an acrylic dimer and Figure 11B shows the same surface after reaction.

This type of treatment results in high water repellency and delays the sorption of water into the wood structure.

Surface hardness can also be increased by compressing (densification) the surface [36,37,38,39,40]. Figure 12 shows the effect of surface densification on specific gravity [38]. The density of densified wood increased up to 1.227 g cm^−1^ which is an 169% increase compared to that of the uncompressed wood [41].

Figure 13 shows the physical change of a wood surface as a result pf compression.

The wood surface can also be laminated, for example, by pressing a layer of heavy-duty melamine containing paper [42,43]. The laminate can have many different types of decorative design. A wood grain pattern can be used but many other designs and colors are available. Figure 14A shows a gray paper impregnated with melamine on the surface of pine.

A decorative wood veneer can be put on the surface of wood (solid or composite) to cover up a low quality board. This practice dates back to ancient Egyptian times when veneers were used on their furniture and sarcophagi. Figure 14B shows a walnut veneer laminated onto the surface of pine.

### 3.6. Adhesives

Adhesive-wood bond performance is influenced by the permeability, surface energy and the depth of penetration of the adhesive into the porous surface [44,45,46,47,48,49,50,51]. A wide variety of adhesives and curing processes are used, but bond strength is based mainly on the physical interaction (entanglement) of the adhesive within the wood surface layers. The ability to “wet” the wood surface is an important factor in adhesive penetration. Figure 15 shows the penetration of a phenolic adhesive on the surface of wood. Bond strength is tested by pulling or shearing apart two test pieces in an ASTM standard test [52].

### 3.7. Coatings

There are many different oil and water based paints and different ways of applying them [53,54,55]. As with adhesives, penetration is generally only into the outer surface cells. The coating is applied to either change color, add protection or both and may include a primer undercoat.

Coatings may be transparent, translucent or opaque, allowing a wide variety of choices. Figure 16 is a microgram of the interface between acrylic paint and a wood surface showing very little penetration into the wood surface.

Coatings fail by peeling, cracking, blistering, and/or flaking mainly due to moisture sorption. Figure 17A shows a surface coating failure due to moisture sorption and Figure 17B shows a coating failure due to end grain moisture sorption [56].

Clear coatings often fail due to the UV transparent properties of the coating [57]. Figure 18 shows a clear coating failure due to ultraviolet energy going through the coating (see Section 3.3).

The adhesion of a coated wood sample of any paint product can be measured by the minimum tensile stress needed to detach the coating perpendicular to the wood surface [58].

Nanotechnology can be used to form a very hard clear coating on a wood surface [59,60]. Nanotechnology is defined as the manipulation of matters between 1 and 100 nm. Nanocoatings on wood use a polymer matrix of mainly nanozinc oxide, nanotitanium oxide or nanosilica that enhance the functionality of the wood surface in terms of durability, hardness, fire resistance and UV absorption as well as decrease in water absorption [61,62,63]. Nanocoatings have recently been used to increase the hardness of wood flooring. Figure 19 shows a wood coated with a nanozincoxide.

Powder and coil coatings are also possible with wood [64].

### 3.8. Fire Protection

An intumescent coating is a fire retardant system which produces gases upon heating that are trapped on the surface of wood. This system acts as a physical barrier to retard both smoldering and flaming combustion by preventing the flammable products from escaping and by preventing oxygen from reaching the substrate [65]. These barriers also insulate the wood from high temperatures. Common barriers include sodium silicates and coatings that intumesce (release a gas at a certain temperature that is trapped in the polymer coating the surface). Intumescent systems swell and char on exposure to fire to form carbonaceous foam and consist of several components. These compounds include a char-producing compound, a blowing agent, a Lewis-acid dehydrating agent, and other chemical components.

Intumescent coatings are commonly used in the construction industry to give improved fire resistance to building materials by reducing the rate of heating and hence prolonging the time for the building materials to reach critical failure temperature.

### 3.9. Mold, Mildew and Fungi

Molds, mildew and fungi include all species of microscopic fungi that grow in the form of multicellular filaments, called hyphae [66,67,68]. All types of molds grow on wood surfaces as well as almost any surface where the mold can survive. There are several common molds that grow in and on wood. One very common one that has been isolated from wood is *Aureobasidium pullulans*. *A. pullulans*, is also called the “black yeast,” which produces a green melanin that turns black over time. Colonies are fast growing, smooth, covered with slimy masses of conidia, brown or black. The mold can use many sources of nutrients to support its growth. Schoeman and Dickerson [68] found that *A. pullulans* can grow on weathered wood surfaces readily utilizing breakdown products from lignin photodegradation as the sole source of carbon and energy (see Section 3.3). These include: extractives in the wood, free sugars, starch, and other available organic compounds. This mold can grow on or through a painted surface. Mold growing on the surface of the paint gets its nutrients passing through the paint film [69,70,71,72] (see Figure 20 and Figure 21).

## 4. Conclusions

I have tried to present a review of the present state of the science of wood surface chemistry. It is not intended to be all inclusive with hundreds of references but present examples of different types of surface chemistry.

Theoretically, if the surface of wood could be protected from moisture, ultraviolet energy, microorganisms, heat, oxygen, and stress, products made from wood would last forever. However, we have never found a way to do that. We can modify the surface with a variety of chemistries to improve water repellency, fungal resistance, hardness, and improve coating and adhesion performance and fire retardancy but all fail, in time, due to several factors.

We need a fresh approach in the study of wood surface chemistry. I suggest we study “dynamic surface envelopes” which implies protecting more than one surface and for more than one application. This implies the surface chemistry is interactive and meant to be more than just a paint or coating.

The surface chemistry could contain micro capsules that deliver a variety of chemicals for water repellency, decay resistance, and/or fire retardancy. The surface chemistry could contain multifunctional bonding sites that could covalently bond two wood surfaces together. The surface chemistry can become “smart” using “quantum dots” so the surface can communicate with us. The dynamic surface can be self-cleaning (Lotus Effect). It can also contain a “performance package” of chemistries intended for more than one application.

## Figures and Tables

**Figure 1 polymers-13-02558-f001:**
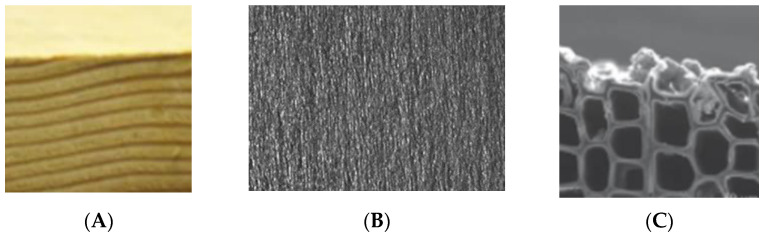
Radiata pine surface (**A**), micrograph wood surface after sanding (**B**), electron-micrograph of wood surface after sanding (**C**).

**Figure 2 polymers-13-02558-f002:**
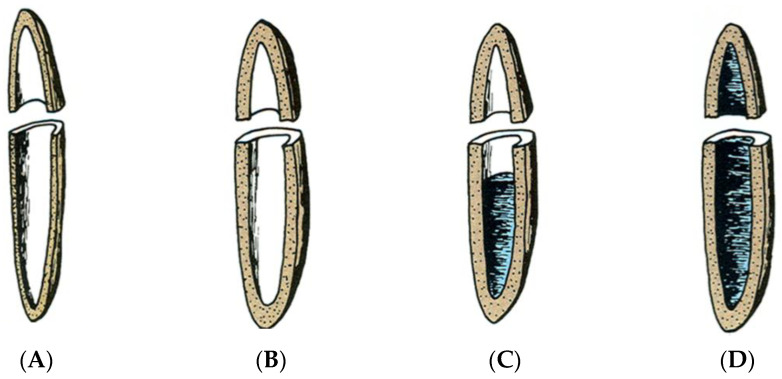
Dry surface fiber (**A**), fiber saturated with moisture (FSP) (**B**), fiber filling with liquid water (**C**), fully saturated fiber (**D**).

**Figure 3 polymers-13-02558-f003:**
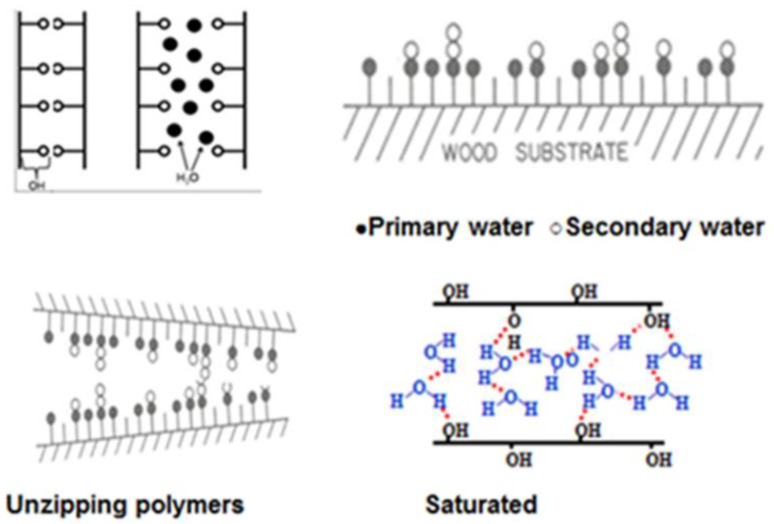
Moisture absorption and adsorption in a wood surface.

**Figure 4 polymers-13-02558-f004:**
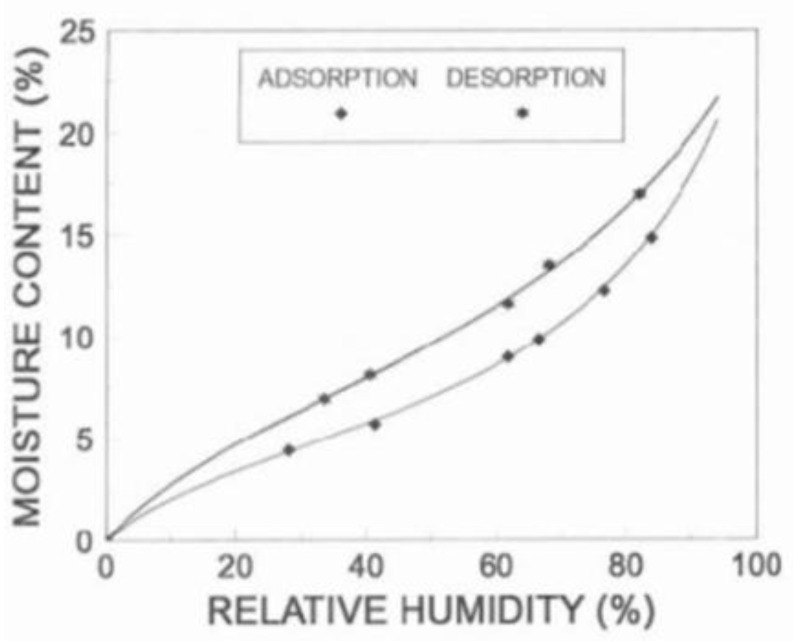
Sorption isotherm for wood.

**Figure 5 polymers-13-02558-f005:**
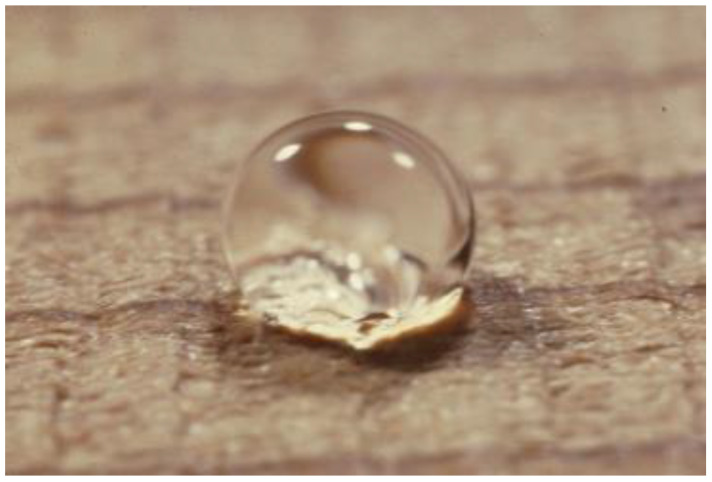
Water droplet on the surface of water repellent treated wood.

**Figure 6 polymers-13-02558-f006:**
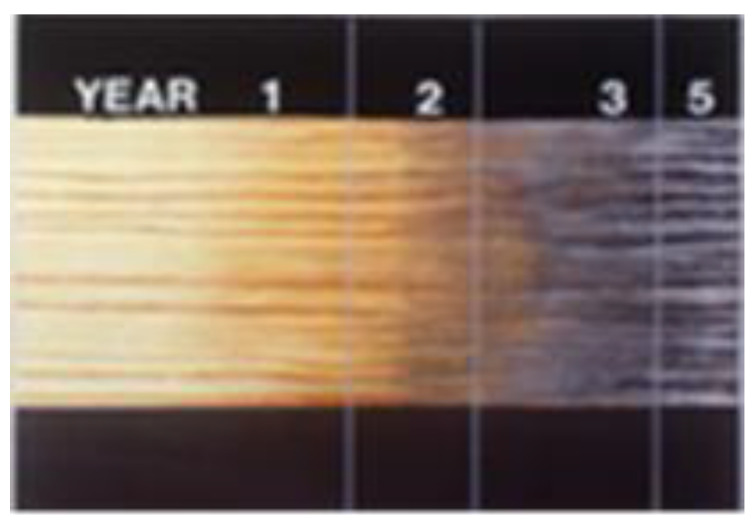
Color change in the wood surface exposed to UV energy.

**Figure 7 polymers-13-02558-f007:**
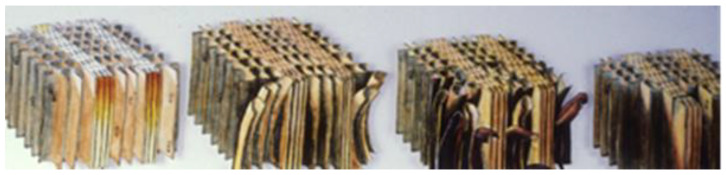
Loss of surface fibers due to UV degradation of lignin.

**Figure 8 polymers-13-02558-f008:**
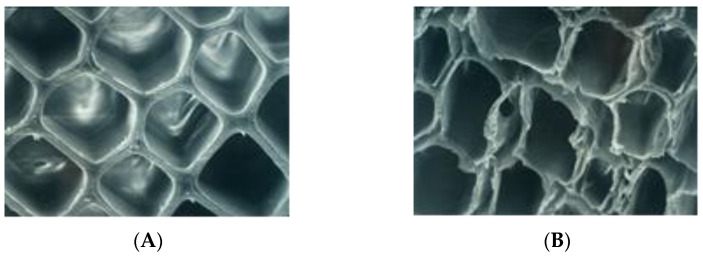
Wood surface before weathering (**A**) and after (**B**).

**Figure 9 polymers-13-02558-f009:**
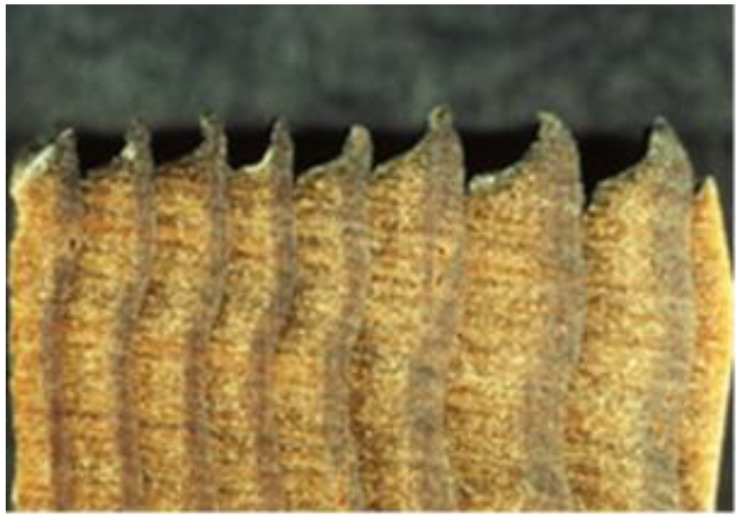
Rough surface of wood after weathering.

**Figure 10 polymers-13-02558-f010:**
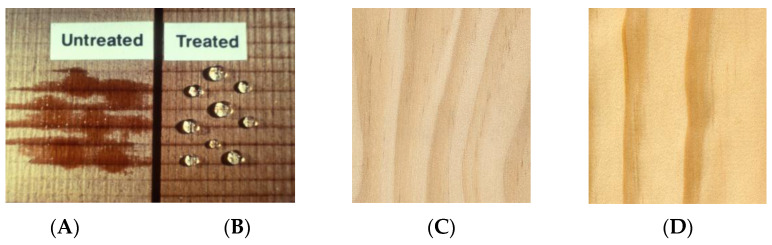
Wood surface before (**A**) and after (**B**) plasma treatment, southern yellow pine before (**C**) and after plasma treatment (**D**).

**Figure 11 polymers-13-02558-f011:**
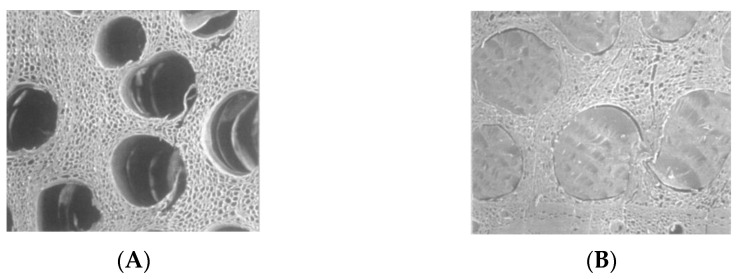
Scanning micrograph of solid wood before polymer impregnation with open lumens (**A**) and after polymer impregnation with filled lumens (**B**).

**Figure 12 polymers-13-02558-f012:**
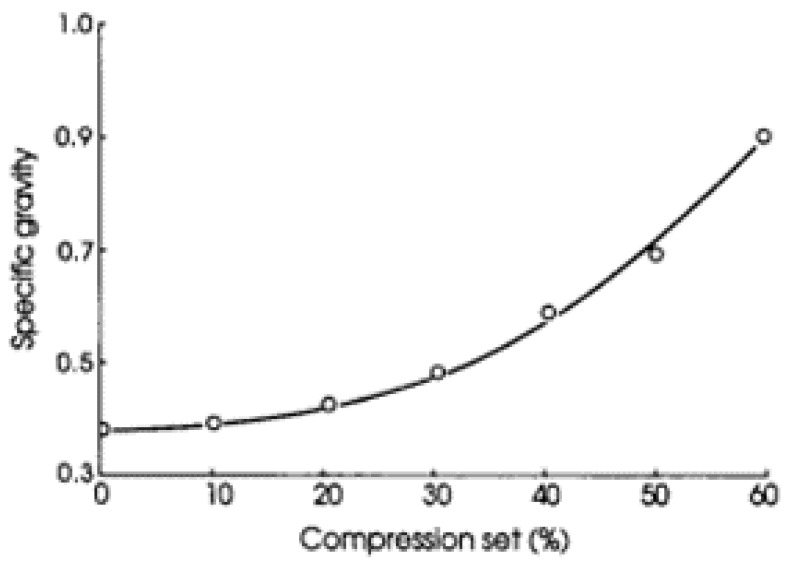
Effect of compression on specific gravity.

**Figure 13 polymers-13-02558-f013:**
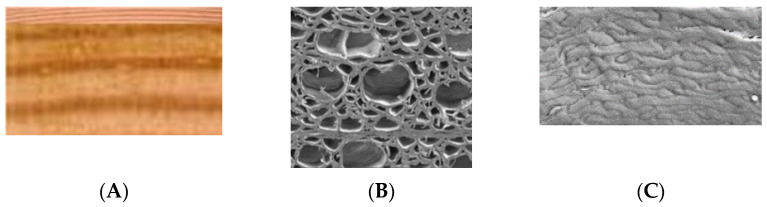
Structure of surface compressed southern pine (**A**) and oak before (**B**) and after compression (**C**).

**Figure 14 polymers-13-02558-f014:**
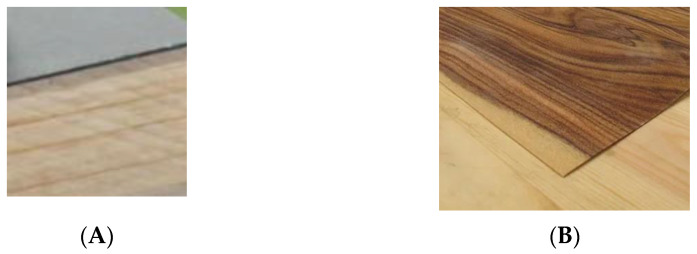
Paper impregnated with melamine on the surface of pine (**A**) and a walnut veneer on the surface of pine (**B**).

**Figure 15 polymers-13-02558-f015:**
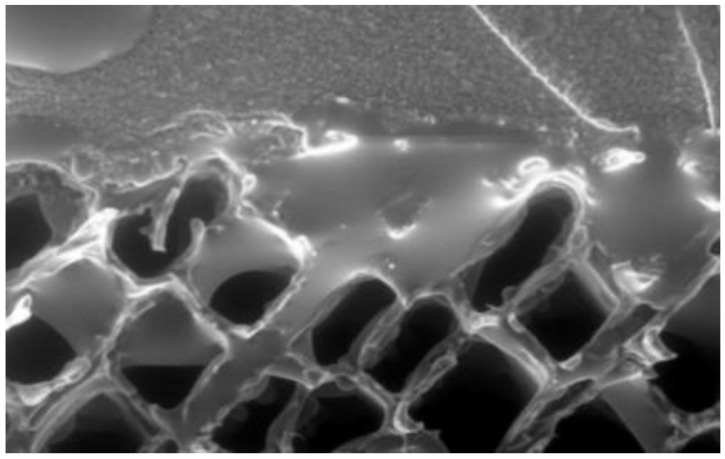
Penetration of an adhesive into the wood surface.

**Figure 16 polymers-13-02558-f016:**
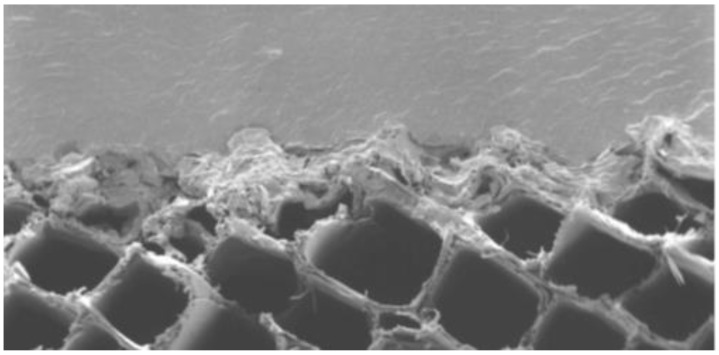
Acrylic paint on the surface of wood.

**Figure 17 polymers-13-02558-f017:**
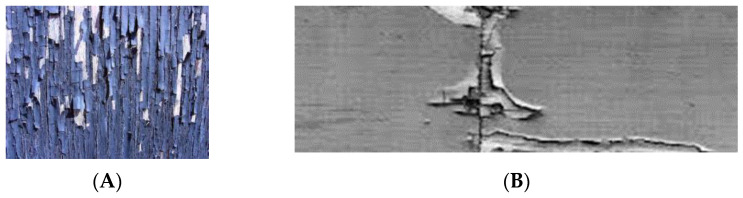
Paint failure due to surface moisture sorption (**A**) and paint failure due to end grain moisture sorption (**B**).

**Figure 18 polymers-13-02558-f018:**
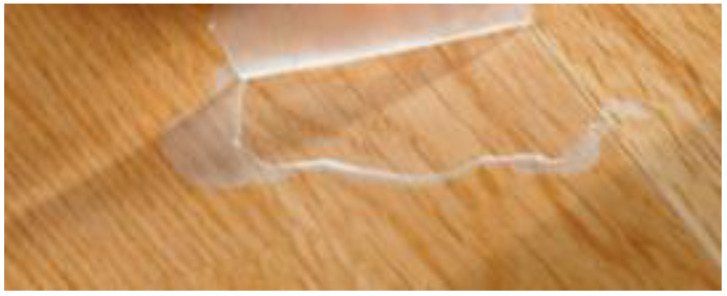
Clear coating failure.

**Figure 19 polymers-13-02558-f019:**
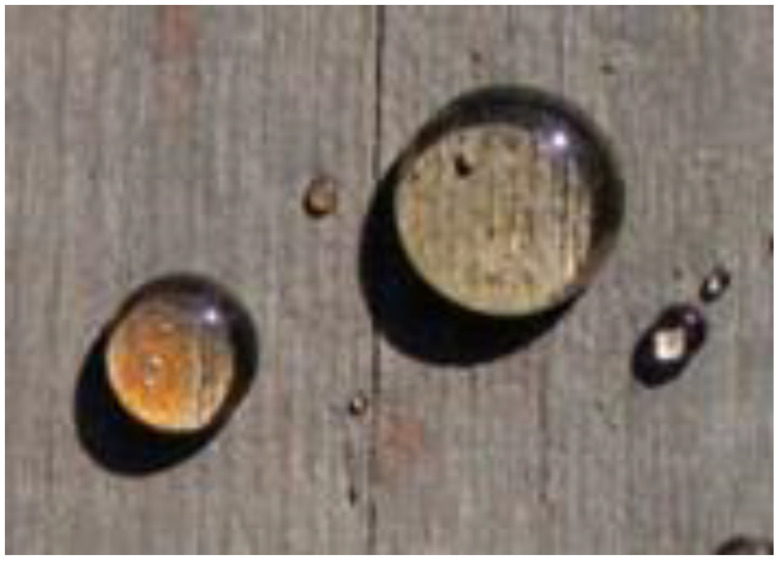
Water droplets on wood with a nanozincoxide coating.

**Figure 20 polymers-13-02558-f020:**
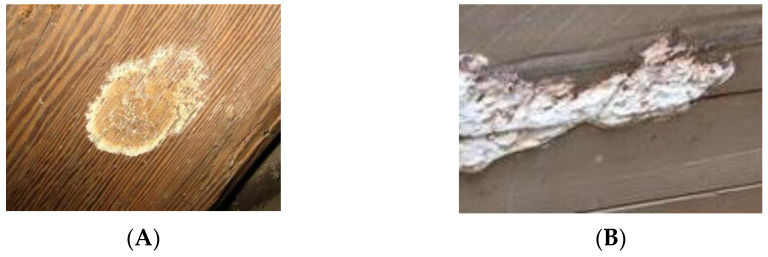
Mold growing on the wood surface (**A**) and mildew on a wood surface (**B**).

**Figure 21 polymers-13-02558-f021:**
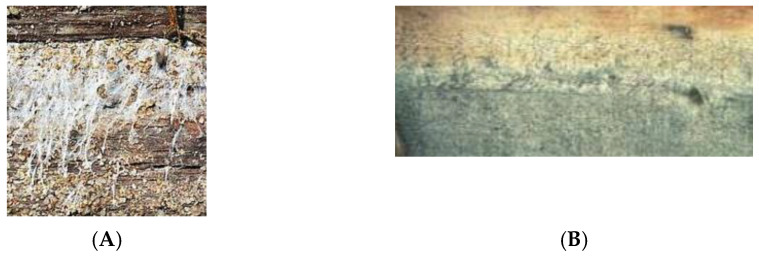
White rot fungi growing on a wood surface (**A**) and mold growing on a painted wood surface (**B**).

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
