# Peer review of "Understanding Wood Surface Chemistry and Approaches to Modification: A Review"

_polymers, 2021, doi:10.3390/polym13152558_

Round 1
Reviewer 1 Report
This manuscript is a good start, but it reads as an introduction to a thesis. It is a reasonable introduction to the terminology used in this area but it does not contribute any new thoughts or insights into the science of wood surface modification.
In my opinion, what is required is a more comprehensive review that discusses the latest developments and how these may improve our understanding of wood surfaces themselves. A listing at the end were the gaps our in the knowledge of wood modification would be a contribution and very helpful for future readers. Without this, the manuscript is just another collection of words to read that does not add anything to the current understanding of the state of the art. .
Author Response
This is a REVIEW not a research manuscript. I agree that it would need a lot more information if it were a research paper. Here is the comment from another reviewer.
This chapter is a review of the chemical properties of a wood surface and performance issues associated with it. This review is very important for wood related scientific research workers. The text is concise and informative, and has important reference value. I recommend publication of this review.
Reviewer 2 Report
This chapter is a review of the chemical properties of a wood surface and performance issues associated with it. This review is very important for wood related scientific research workers. The text is concise and informative, and has important reference value. I recommend publication of this review.
Comment: The quality (resolution) of figure needs to be improved in the manuscipt.
Author Response
I se that no changes are needed from this reviewer.